# Cohort profile: Data standards for cardiac rehabilitation structure and processes for the SWEDEHEART cardiac rehabilitation (SWEDEHEART-CR) registry

Margret Leosdottir[1,2]*, Maria Bäck[3,4], Lars Dahlbom[5,6], Mattias Ekström[7], Bertil Lindahl[6,8], Emil Hagström[6,8]

1 Department of Clinical Sciences Malmö, Faculty of Medicine, Lund University, Malmö, Sweden, 2 Department of Cardiology, Skane University Hospital, Malmö, Sweden, 3 Division of Physiotherapy, Department of Medical and Health Sciences, Linköping University, Linköping, Sweden, 4 Department of Occupational Therapy and Physiotherapy, Sahlgrenska University Hospital, Gothenburg, Sweden, 5 Department of Cardiology, Bollnäs Hospital, Bollnäs, Sweden, 6 Uppsala Clinical Research Centre, Uppsala, Sweden, 7 Division of Cardiovascular Medicine, Department of Clinical Sciences, Karolinska Institutet, Danderyd Hospital, Stockholm, Sweden, 8 Department of Medical Sciences, Cardiology, Uppsala University, Uppsala, Sweden

* margret.leosdottir@med.lu.se

## Abstract

Data standards for quality registries should be evidence-based and follow guideline recommendations. To optimally monitor quality of care, not only patient-level variables, but also centre-level variables need to be included. Here we describe the development of variables to audit the structure and processes in cardiac rehabilitation for patients after myocardial infarction, and the resulting data standards to be implemented in the Swedish quality registry for cardiac disease, SWEDEHEART. The methodology used for the development of international clinical data standards for the European Unified Registries for Heart Care Evaluation and Randomised Trials (EuroHeart) was followed. Based on national guidelines for secondary prevention, candidate variables were prepared, after which a multiprofessional expert group on cardiac rehabilitation selected key variables and assured face validity. An external reference group had the role of peer reviewing, ascertaining content validity and test-retest reliability. The process has resulted in 30 data standards to be introduced into the SWEDEHEART cardiac rehabilitation registry and administered on centre-level biannually. The data standards include measures of human resources, centre requirements and process-based metrics. Including registry variables which audit centre-level structure and processes is essential to improve benchmarking and standardize monitoring of quality of care, covering both services provided and patient outcomes.

## Introduction

Interventions targeted at cardiovascular risk factors and the fostering of a healthy lifestyle after an acute myocardial infarction (MI), i.e., secondary prevention, is the most effective tool to

**Data Availability Statement:** All relevant data are within the manuscript and its Supporting Information files.

**Funding:** The authors received no specific funding for this work.

**Competing interests:** The authors have declared that no competing interests exist.

reduce recurrent cardiovascular events [1]. Administering secondary prevention via structured cardiac rehabilitation (CR) programmes for patients with chronic coronary syndromes (CCS) reduces mortality, morbidity, and improves quality of life [2, 3]. Comprehensive CR is a complex multidisciplinary intervention, administered by a team and usually coordinated by a cardiologist. It comprises patient assessment, management of cardiovascular risk factors, including optimal use of cardio-protective pharmacotherapy, exercise training, behavioral modification, patient education, lifestyle and psychosocial counselling and vocational support [4]. Even though the optimal delivery of CR has been defined [4, 5] there is considerable heterogeneity in which components are included in CR programmes within and between countries, which in turn could affect both adherence to pharmacotherapy and healthy lifestyle advice as well as treatment target attainment [1, 4, 6–9].

Continuous monitoring of quality of care is pivotal to identify gaps between guideline-recommended interventions and actual care in clinical practice [10, 11]. The Swedish Web-system for Enhancement and Development of Evidence-based care in Heart disease Evaluated According to Recommended Therapies (SWEDEHEART) is a nationwide quality registry that records characteristics, treatments during the acute phase and follow-up, and outcomes of consecutive patients with MI admitted to coronary care units in Sweden [12, 13]. Most variables in the CR part of the registry (SWEDEHEART-CR) audit patient-level outcomes (pharmacotherapy, lifestyle, risk factors, patient-reported outcome measures) and a few variables audit CR service delivery (patient participation in and timing of individual CR programme components).

Implementing evidence-based medicine is a complex process faced with many challenges, and in far too many instances, guideline recommendations result in little or no change in clinical practice [14, 15]. Being nationwide with more than 80 percent patient coverage at one-year post MI, the SWEDEHEART registry offers a unique possibility to monitor guideline implementation on a national level, benchmarking and quality control of the structure and processes applied within CR, as recommended in guidelines. For this purpose, to be able to measure CR quality, key variables to audit structure and processes recommended in recently published National Guidelines on Secondary Prevention for patients with CCS in Sweden [16] have been proposed by the SWEDEHEART-CR Working Group.

Here we describe the development process of variables to audit the structure and processes recommended in National Guidelines on Secondary Prevention for patients with CCS and resulting data standards to be implemented in the SWEDEHEART-CR registry.

## Cohort description

### Development of data standards

When deciding on the data standards, the methodology used for the development of international clinical data standards for the European Unified Registries for Heart Care Evaluation and Randomised Trials (EuroHeart) was followed [17]. The domains covered were the structure and processes applied within CR, as recommended in the National Guidelines on Secondary Prevention for patients with CCS [16]. The process of producing the data standards is summarized in Fig 1.

A Data Science Group comprised of three members from the SWEDEHEART-CR Working Group (M.L., M.B. and E.H.). M.L. had the role of project manager. The Data Science Group prepared a list of candidate variables (S1 Table). The candidate variables were based on recommendations in the National Guidelines [16]. The National Guidelines were in turn based on the European Society of Cardiology (ESC) Guidelines on cardiovascular disease prevention in clinical practice [2]; diabetes, pre-diabetes, and cardiovascular diseases [18]; the diagnosis and

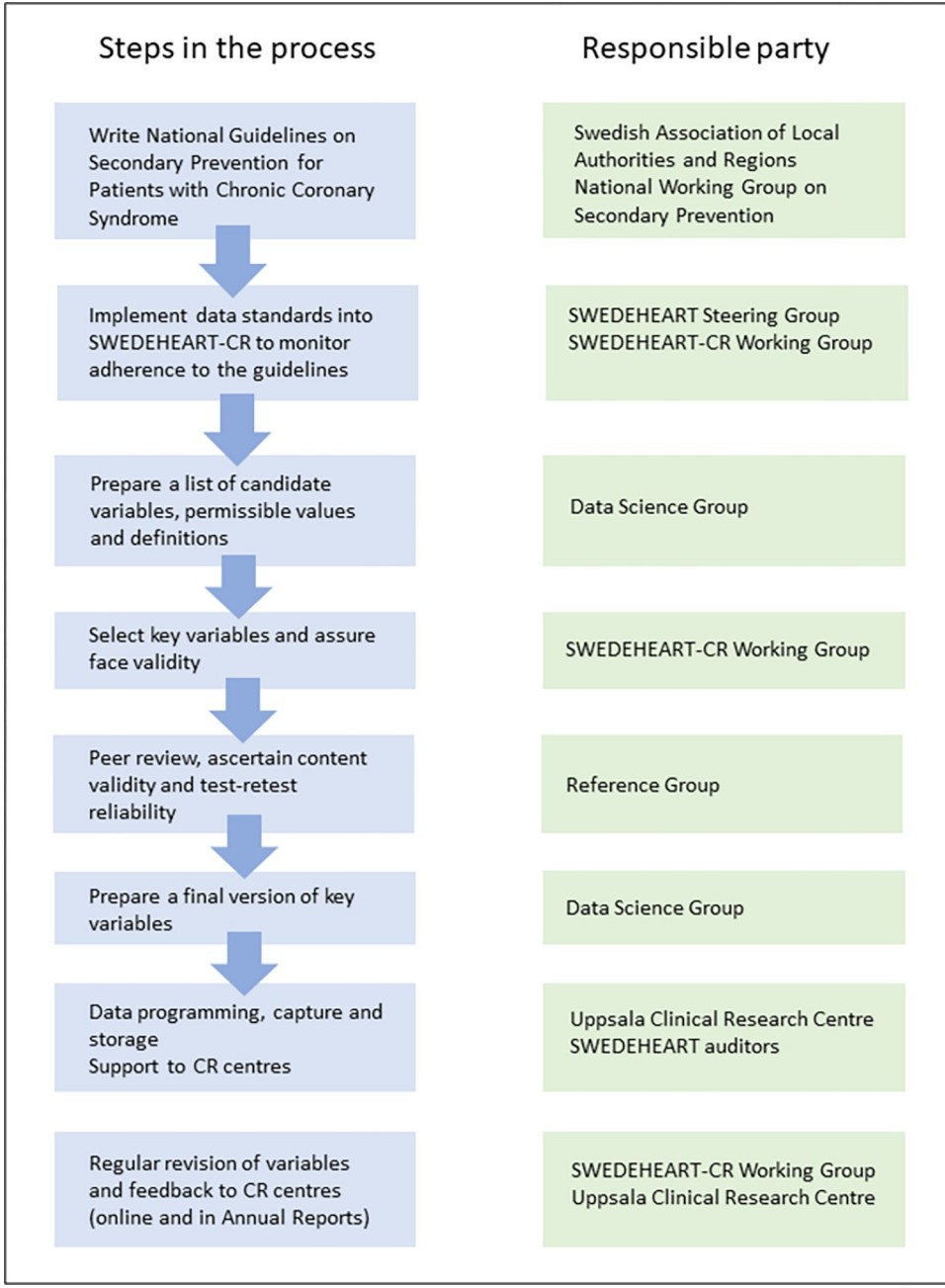

**Fig 1. Schematic diagram of the development process for constructing data standards for cardiac rehabilitation processes and structures within the SWEDEHEART-CR registry.** CR, cardiac rehabilitation; SWEDEHEART, The Swedish Web-system for Enhancement and Development of Evidence-based care in Heart disease Evaluated According to Recommended Therapies.

management of CCS [19]; and the management of arterial hypertension [20], adapted to the structure of CR as provided in Sweden. Standards in the National Guidelines were also defined in accordance with recommendations on standardization and quality improvement of secondary prevention through CR programmes in Europe from the European Association of Preventive Cardiology (EAPC) [4, 5]. Variables covering central recommendations which were already implemented at all or close to all (>95%) CR centres in Sweden, were removed at this

stage [21]. These included for example assessment of dietary habits, level of physical activity, and tobacco use being included in routine follow-up, and that the CR centre reports to a quality registry (S1 Table).

The Data Science Group also suggested permissible values for the variables. Three values were suggested: Yes, Partly and No. The value Partly (Swedish "Delvis") was included to reduce the risk of a ceiling effect with dichotomous answers. For the human resources domain, the Partly value enables a more nuanced answer than Yes/No if the CR team only has a part-time service from one of the professions mentioned. This mostly applies to dieticians, psychologists, and social workers. For the other variables this permissible value denotes if the routine is partly implemented/followed (i.e., using registry data continuously for improving quality of care) or if a part of the staff has the respective qualification (i.e., some of the nurses in the CR team having delegation to titrate medication).

The SWEDEHEART-CR Working Group had the role of selecting key variables and assuring face validity. The working group comprised 13 members: six cardiologists, one primary care physician, two physiotherapists, two nurses, and two psychologists. All members have extensive clinical and/or research experience within CR and between 1–12 years of experience working with quality registries. Candidate variables were presented by the Data Science Group to the Working Group which were asked to select key variables based on the following criteria:

- The variable is included in the National Guidelines

- The variable is included in ESC guidelines or EAPC consensus documents

- The variable is associated with patient outcomes

- The data standard has room for improvement within Swedish CR

- The variable is clearly written and objective

- Clear format and structure of the information to be captured within a variable (permissible value)

To be accepted as key variables, candidate variables should fulfil at least 4 out of these 6 criteria. The Working Group came to a joint conclusion after three discussion rounds (two e-mail discussion rounds and one face-to-face meeting). Phrasing was further adapted after the final discussion round.

A Reference Group was comprised of six cardiologists, ten nurses, and five physiotherapists from eight CR centres in Sweden. The Reference Group had the role of peer reviewing, ascertaining content validity and test-retest reliability. The group had a wide geographical representation and included representatives from small (less than 50 post MI patients followed per year, n = 3), medium-sized (50–150 patients followed per year, n = 3), and large (more than 150 patients followed per year, n = 2) CR centres across the country. Additionally, five nurses specialized in cardiac care who work for the SWEDEHEART registry as service contacts for registry users were included in the Reference Group. Key variables were reviewed by the Reference Group, providing critical feedback on the selection of variables (using the same criteria as listed above), as well as applicability and generalizability. The final set of key variables can be seen in Table 1. Process variables already included in the SWEDEHEART-CR registry are shown in Table 2.

## Setting and populations

The SWEDEHEART-CR registry was started in 2005 [12]. Since 2018 all CR centres in the country report data to the registry (n = 78). All CR centres reporting to SWEDEHEART-CR will be subject to the new data standards. The patient population included in

**Table 1. New data standards for auditing structure and processes in cardiac rehabilitation.**

| Variable | Permissible values | | | |
|---|---|---|---|---|
| **_Structure-based metrics_** | | | | |
| _Human resources_ | | | | |
| The following professions are included in our CR team: | | | | |
| Nurse | Yes | Partly | No | Unknown |
| Physiotherapist | Yes | Partly | No | Unknown |
| Physician | Yes | Partly | No | Unknown |
| Social worker | Yes | Partly | No | Unknown |
| Psychologist | Yes | Partly | No | Unknown |
| Dietician/nutritionist | Yes | Partly | No | Unknown |
| The CR centre has a medical director with cardiology training who is responsible for the oversight of programme policies and medical procedures | Yes | Partly | No | Unknown |
| Nurses at the CR centre have an individual delegation to titrate dosage/suggest changes in lipid lowering therapy | Yes | Partly | No | Unknown |
| Nurses at the CR centre have an individual delegation to titrate dosage/suggest changes in blood pressure lowering therapy | Yes | Partly | No | Unknown |
| Personnel at the CR centre has training in counselling methods (e.g., motivational interviewing or cognitive behavioural therapy) | Yes | Partly | No | Unknown |
| At least one member of the CR team has training in tobacco counselling | Yes | Partly | No | Unknown |
| _Centre requirements_ | | | | |
| We have regular interdisciplinary team meetings to discuss patient cases | Yes | Partly | No | Unknown |
| We have regular interdisciplinary team meetings to discuss operational matters such as work routines, quality of care, and to improve the team spirit | Yes | Partly | No | Unknown |
| We use SWEDEHEART-CR data continuously to follow and improve quality of care at our CR centre | Yes | Partly | No | Unknown |
| **_Process-based metrics_** | | | | |
| We follow and act on the patients´ identified modifiable risk factors | Yes | Partly | No | Unknown |
| We follow and act on the patients´ adherence to and effect of pharmacological treatment | Yes | Partly | No | Unknown |
| We strive for continuity in patient-caretaker contact throughout follow-up | Yes | Partly | No | Unknown |
| We offer the patients´ relatives to attend follow-up visits | Yes | Partly | No | Unknown |
| For non-Swedish speaking patients, certified interpreter services are used | Yes | Partly | No | Unknown |
| Nicotine-replacement therapy is offered to smokers | Yes | Partly | No | Unknown |
| Bupropion, cytisin or varenicline therapy is offered to smokers | Yes | Partly | No | Unknown |
| Assessment of alcohol consumption is included in routine follow-up | Yes | Partly | No | Unknown |
| We offer participation in supervised exercise-based CR for at least 3 months (24 sessions) | Yes | Partly | No | Unknown |
| For patients with high office blood pressure, we measure home and/or ambulatory blood pressure | Yes | Partly | No | Unknown |
| Fasting glucose and HbA1c are controlled during follow-up for all patients | Yes | Partly | No | Unknown |
| When fasting glucose and/or HbA1c are inconclusive, oral glucose tolerance test is performed | Yes | Partly | No | Unknown |
| Our cardiologists initiate and optimize treatment for type-2 diabetes | Yes | Partly | No | Unknown |
| Psychosocial status assessment and management (if needed) are included in routine follow-up | Yes | Partly | No | Unknown |
| Vocational counselling and support are included in routine follow-up | Yes | Partly | No | Unknown |
| We offer interactive patient education as a part of the CR programme | Yes | Partly | No | Unknown |

CR, cardiac rehabilitation; HbA1c, haemoglobin A1c; MI, myocardial infarction.

SWEDEHEART-CR has been previously described in detail [12]. In short, all patients with a type-1 MI discharged alive from coronary care units around the country are eligible for CR follow-up and are consequently eligible for registration in SWEDEHEART-CR.

## Ethical considerations

The need for signed informed consent by patients for inclusion in Swedish quality registries has collectively been waived in Sweden. At hospital admission, MI patients are informed verbally and in writing about data being collected and entered in the registry. All patients have the

**Table 2. Data variables already included in the SWEDEHEART-CR registry auditing cardiac rehabilitation processes.**

| Variable | Permissible values |
|---|---|
| *Attendance in CR programme* | |
| Proportion of patients attending an initial CR assessment (nurse visit) | Proportion (%) |
| Proportion of patients attending an individual visit to a physiotherapist post-discharge before starting an EBCR programme (a pre-exercise assessment visit) | Proportion (%) |
| Proportion of patients attending an EBCR programme | Proportion (%) |
| Proportion of patients attending an individual close-out visit to a physiotherapist after completing an EBCR programme (a post-exercise assessment visit) | Proportion (%) |
| Proportion of patients attending interactive patient education | Proportion (%) |
| Proportion of patients attending a close-out CR visit with a nurse | Proportion (%) |
| *Time to start in different components of CR* | |
| Time from discharge to initial CR assessment (nurse visit) | Days (n) |
| Time from discharge to pre-exercise assessment (physiotherapist visit) | Days (n) |
| Time from discharge to start in EBCR programme | Days (n) |

CR, cardiac rehabilitation; EBCR, exercise-based cardiac rehabilitation.

right to deny registration and upon request to be removed from the registry at any time. As the data standards described herein will be reported at centre-level there is no need for additional informed consent at patient-level or ethical review board consent.

In an upcoming trial, however, we aim to evaluate whether auditing structure and processes of CR through the SWEDEHEART registry can increase adherence to national guidelines and improve patient outcomes. This trial has been evaluated and approved by the Swedish Ethical Review Board (Dnr 2023-03217-01; ClinicalTrials.gov ID: NCT 05889416).

## Baseline and follow-up data

Patient-level data from consecutive patient follow-up visits at the CR centre is reported at fixed time points after the MI to SWEDEHEART-CR, with data collected from electronic medical records (i.e., comorbidities, current medication, and re-admissions), self-report (i.e., symptoms, physical activity, and dietary habits), or by measurement (i.e., blood pressure, lipid levels) (Fig 2). As the data standards proposed here are not on a patient-level, they will be administered on centre-level biannually. The variables will be accessible for one month at a time. To receive responses from all key CR professions (nurse, physiotherapist, and physician), the CR team will be encouraged to answer the questions together. The data standards will be launched during the fall 2023. To acquire a baseline status all CR centres will complete the questions for the first time. Thereafter, to enable evaluation of whether auditing service delivery can alter implementation of the national guidelines, centers will be randomized to report data bi-annually for three years or report no data.

## Data capture and storage

SWEDEHEART-CR data is collected through an interactive web-based platform developed and maintained by Uppsala Clinical Research Centre (UCR; http://www.ucr.uu.se). The data is entered online by healthcare professionals at each CR centre. Patient-level data is transferred encrypted and stored in a central secure server. Data for benchmarking is continuously accessible online and is published annually in the SWEDEHEART annual reports (http://www.ucr.uu.se/swedeheart).

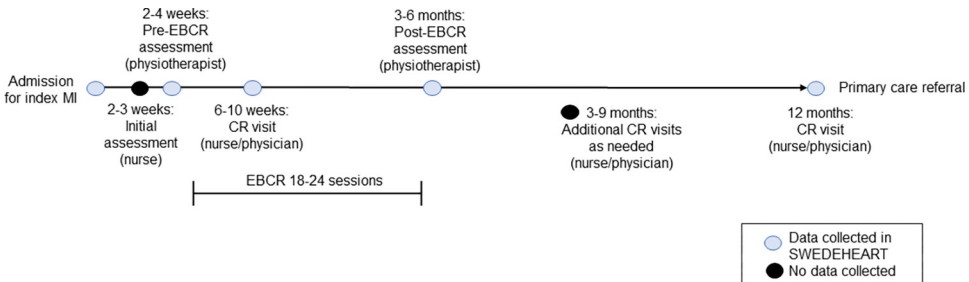

**Fig 2. The basic structure of visits within cardiac rehabilitation (CR) as applied at most CR centres in Sweden.**
Visits at which data is collected in the SWEDEHEART registry are marked in light blue. CR, cardiac rehabilitation;
EBCR, exercise-based cardiac rehabilitation; MI, myocardial infarction.

## Data quality, endpoints, and linkage to other data

As the primary aim of the data standards is to audit implementation of national guidelines, no
systematic literature review was performed when preparing the candidate variables. In some
instances, when converting a recommendation from the guidelines to a candidate variable, the
phrasing was adapted for clarity and to increase objectiveness.

Data entry to SWEDEHEART is voluntary, and centres get no reimbursement for doing so.
By entering data, however, the centres and their patients have continuous online access with-
out direct cost to local and national benchmarking. As for all other data entered to SWEDE-
HEART, a user manual with definitions of each variable will be provided to ensure correct
registration of data. Also, the service contacts are available to aid healthcare personnel respon-
sible for data entry if needed. The data platform has error checking for range and consistency.
Since the start of the registry, SWEDEHEART data has been internally audited every other
year to certify the correctness of the data entered. The auditing has consistently confirmed
>95% agreement with data from medical records [12].

Since every patient for whom data is entered into the SWEDEHEART registry has a unique
personal identification number and is linked to the CR centre responsible for their follow-up,
the data standards described here can be linked to patient data at each centre. This allows
auditing not only of adherence to guidelines (data standards proposed) but also patient out-
comes. Linkage with other nationwide registries in Sweden which collect data on hospital
admissions, diagnoses, causes of death, pharmacotherapy, and socioeconomic data can also be
performed using the personal identification number.

## Findings to date

To assure test-retest reliability the Reference Group members were in December 2022 asked to
complete the questionnaire twice with a one-month interval. All answers were collected by
February 2023. Using Cohen´s weighted kappa the test-retest reliability was high (k = 0.930,
confidence interval 0.870–0.984, p<0.0001).

Responses to the structure and process variables provided by the eight CR centres repre-
sented in the Reference Group are shown in Fig 3.

## Strengths and limitations

A major strength of the cohort is that the data standards described here will be introduced in
the SWEDEHEART registry, which has 100% national coverage on CR centre-level. All hospi-
tals and CR centres entering data into SWEDEHEART have direct access to their own data, to
be used for benchmarking and local quality improvement. Also, the data standards have been

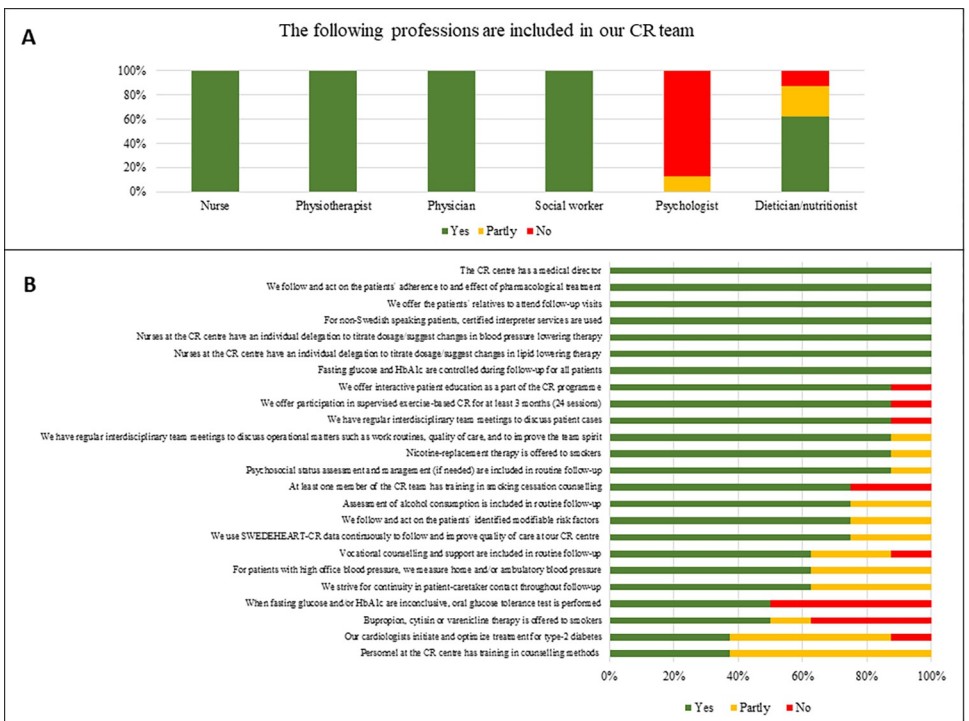

**Fig 3. Responses provided by the eight CR centres represented in the Reference Group.** The upper panel (A) shows composition of the CR team, and the lower panel (B) shows the remaining variables in falling order of proportion reported as yes. CR, cardiac rehabilitation; EBCR, exercise-based cardiac rehabilitation; HbA1c, haemoglobin A1c; MI, myocardial infarction.

developed using an acknowledged methodology and are based on CR structure and processes as recommended in national and international guidelines on secondary prevention for patients with CCS. A major limitation to this work is that no systematic literature review was performed when preparing the candidate variables. This was however deemed as an unnecessary step as the data standards are based on clinical recommendations as presented in international guidelines.

## Future plans

Whether auditing service delivery of CR within a national quality registry can improve adherence to guidelines and patient outcomes has to our knowledge not been scientifically evaluated. Therefore, as a next step we aim to evaluate whether auditing structure and processes of CR in a national registry can increase adherence to national guidelines and improve patient outcomes.

## Supporting information

**S1 Dataset. Data collected from the Reference group for assuring test-retest reliability.**
(XLSX)

**S1 Table. A list of candidate variables.**
(PDF)

## Acknowledgments

We would like to thank the SWEDEHEART-CR Working Group members, the SWEDE-HEART auditors, and members of the Reference Group for help with generating the data standards.

## Author Contributions

**Conceptualization:** Margret Leosdottir, Maria Bäck, Emil Hagström.

**Data curation:** Margret Leosdottir.

**Formal analysis:** Margret Leosdottir.

**Investigation:** Margret Leosdottir.

**Methodology:** Margret Leosdottir, Maria Bäck, Lars Dahlbom, Mattias Ekström, Emil Hagström.

**Project administration:** Margret Leosdottir.

**Supervision:** Bertil Lindahl, Emil Hagström.

**Validation:** Lars Dahlbom, Mattias Ekström.

**Visualization:** Margret Leosdottir, Maria Bäck, Emil Hagström.

**Writing – original draft:** Margret Leosdottir.

**Writing – review & editing:** Maria Bäck, Lars Dahlbom, Mattias Ekström, Bertil Lindahl, Emil Hagström.

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
