## [Decision Letter · Decision Letter 0]

22 Aug 2023

PONE-D-23-16424Cohort profile: Data standards for cardiac rehabilitation structure and processes for the SWEDEHEART cardiac rehabilitation (SWEDEHEART-CR) registryPLOS ONE

Dear Dr. leosdottir,

Thank you for submitting your manuscript to PLOS ONE. After careful consideration, we feel that it has merit but does not fully meet PLOS ONE’s publication criteria as it currently stands. Therefore, we invite you to submit a revised version of the manuscript that addresses the points raised during the review process.

The reviewers unanimously rate your manuscript very positively overall. The comments focus mainly on transparency. I am sure you can easily implement the definitions and descriptions requested.

We look forward to receiving your revised manuscript.

Kind regards,

Annett Salzwedel

Academic Editor

PLOS ONE

Reviewers' comments:

Reviewer's Responses to Questions

**Comments to the Author**

1. Is the manuscript technically sound, and do the data support the conclusions?

Reviewer #1: Yes

Reviewer #2: Yes

2. Has the statistical analysis been performed appropriately and rigorously? 

Reviewer #1: N/A

Reviewer #2: N/A

3. Have the authors made all data underlying the findings in their manuscript fully available?

Reviewer #1: Yes

Reviewer #2: Yes

4. Is the manuscript presented in an intelligible fashion and written in standard English?

Reviewer #1: Yes

Reviewer #2: Yes

5. Review Comments to the Author

Reviewer #1: The manuscript describes the development of variables to audit the quality of cardiac rehabilitation and data standards in Sweden. It is well written, but some topics need to be added:

- the list of candidate variables and selected key variables (page 5),

- the source for variables associated with patient outcomes;

- it should be explained why only 4 of 6 criteria for acceptance as key variable were choosen and which criteria were used how often per candidate variable,

- a definition of "small", "medium" and "large" CR centers is necessary (page 6),

- in table 1 the definiton of "partly" for the parameters like human resources / centre requirements / process based metrics is necessary and may be different for the particular parameters;

- basis of fig 3 are 8 centers (page 11), therefore it is unclear why results like ~35 (Personnel at the CR centre has training in counselling methods) are possible.

Reviewer #2: This manuscript describes data standards registry for cardiac rehab after AMI with focus on centre-level variables in addition to patient level variables. The data standards will be inplemented to the excellent Swedish quality registry for cardiac diseases. The paper iw well written and give useful insight on the process of choosing the variables according to current guidelines and scientific statements that can be copied by other registries in other countres. The systematic collection of the suggested data will probalbe imrove and standardize the quality of care.

I have just som very minor comments:

I am surprised that the teams consist of psychologist (not so easily avaiable in many places, and much of the psychological aspects could be covered by nurse and physician) and not by pharmacists, but thie paper certaily does not aim to suggest how the teams are put together.

In table 1 I would suggest to delete the category Prtly for the human resources (what is a partly nurse?)

In table 1 I would suggest to define follow-up visit (when, how many) and what is it, how comprehensive

I also think that CR programme (as for example in table 2) should be defined

Settings and populations: I sthis CR registry (and program) only for patients after an MI. WHat about patients with stable coronary disase but whitout an MI, only PCI/CABG? Do they have separate CR programs?

6. PLOS authors have the option to publish the peer review history of their article (what does this mean?). If published, this will include your full peer review and any attached files.

Reviewer #1: No

Reviewer #2: No

---

## [Author Response · Author response to Decision Letter 0]

6 Oct 2023

Responses to reviewers and editor are found in the attached file "Response to reviewers".

---

## [Decision Letter · Decision Letter 1]

20 Oct 2023

Cohort profile: Data standards for cardiac rehabilitation structure and processes for the SWEDEHEART cardiac rehabilitation (SWEDEHEART-CR) registry

PONE-D-23-16424R1

Dear Dr. leosdottir,

We’re pleased to inform you that your manuscript has been judged scientifically suitable for publication and will be formally accepted for publication once it meets all outstanding technical requirements.

Kind regards,

Annett Salzwedel

Academic Editor

PLOS ONE

Additional Editor Comments (optional):

Reviewers' comments:

Reviewer's Responses to Questions

**Comments to the Author**

1. If the authors have adequately addressed your comments raised in a previous round of review and you feel that this manuscript is now acceptable for publication, you may indicate that here to bypass the “Comments to the Author” section, enter your conflict of interest statement in the “Confidential to Editor” section, and submit your "Accept" recommendation.

Reviewer #1: (No Response)

Reviewer #2: All comments have been addressed

2. Is the manuscript technically sound, and do the data support the conclusions?

Reviewer #1: (No Response)

Reviewer #2: Yes

3. Has the statistical analysis been performed appropriately and rigorously? 

Reviewer #1: (No Response)

Reviewer #2: I Don't Know

4. Have the authors made all data underlying the findings in their manuscript fully available?

Reviewer #1: (No Response)

Reviewer #2: Yes

5. Is the manuscript presented in an intelligible fashion and written in standard English?

Reviewer #1: (No Response)

Reviewer #2: Yes

6. Review Comments to the Author

Reviewer #1: (No Response)

Reviewer #2: Very good answers to my review comments as well as the othe reviewer's comments. This paper is important to publish.

Studies on cardiac rehabilitation are few, and this study adds to the literature with good quality

I have no more suggestions.

I think the paper is a good example for other researchers in this area

7. PLOS authors have the option to publish the peer review history of their article (what does this mean?). If published, this will include your full peer review and any attached files.

Reviewer #1: No

Reviewer #2: **Yes: **Maja-Lisa Løchen

---

## [Editor Report · Acceptance letter]

27 Oct 2023

PONE-D-23-16424R1 

Cohort profile: Data standards for cardiac rehabilitation structure and processes for the SWEDEHEART cardiac rehabilitation (SWEDEHEART-CR) registry 

Dear Dr. leosdottir:

I'm pleased to inform you that your manuscript has been deemed suitable for publication in PLOS ONE. Congratulations! Your manuscript is now with our production department. 

Kind regards, 

on behalf of

Dr. Annett Salzwedel 

Academic Editor

PLOS ONE